# Polybasic Speculative Decoding Under a Theoretical Perspective

## Abstract

Speculative decoding has emerged as a critical technique for accelerating inference in large language models, achieving significant speedups while ensuring consistency with the outputs of the original models. However, there is currently a lack of theoretical guidance in speculative decoding. As a result, most existing works are dualistic target-draft model paradigm, which significantly restricts the hinders potential application scenarios. In this paper, we propose a polybasic speculative decoding framework supported by a solid theoretical foundation. We first deduce a theorem to control the ideal inference time of speculative decoding systems which is then serve as a design criterion that effectively expands the original dualistic speculative decoding into a more efficient polybasic speculative decoding. We further theoretically analyze the sampling process, identifying variables that can be optimized to enhance inference efficiency in multi-model systems. We demonstrate, both theoretically and empirically, that this system accelerates inference for the target model, and that our approach is orthogonal to the majority of existing speculative methods, allowing for independent application or combination with other techniques. Experimentally, we conducted comprehensive evaluations across a wide range of models, including those from the Vicuna, LLaMA2-Chat, and LLaMA3 families. Our method achieved remarkable latency speedup ratios of $3.31\times$-$4.01\times$ for LLaMA2-Chat 7B, up to $3.87\times$ for LLaMA3-8B, and up to $4.43\times$ for Vicuna-7B, while maintaining the distribution of the generated text. Code is available in supplementary materials.

## 1 Introduction

Large Language Models (LLMs) have become the core driving force in the field of natural language processing (NLP), demonstrating remarkable performance in various applications. However, the scale and complexity of these models also bring significant computational challenges, especially in real-time application scenarios. Inference acceleration has become a key issue in deploying and applying these models. Among numerous acceleration techniques, speculative decoding(Stern et al., 2018) (Leviathan et al., 2023) has emerged as a critical technique, gaining widespread application in large-scale model deployment.

In recent years, the field of NLP has witnessed significant advancements in speculative sampling methods, leading to the emergence of a "draft-then-verify" paradigm. This approach encompasses various drafting strategies, such as the utilization of small-scale draft models to facilitate speculative sampling in LLMs (Leviathan et al., 2023) (Xia et al., 2023a)(Chen et al., 2023a) (Kim et al., 2024) (Svirschevski et al., 2024), the implementation of tree structures to organize tokens generated by draft models (Miao et al., 2024) (Du et al., 2024) (Stern et al., 2018), the employment of unified models serving as both draft and target models (Yi et al., 2024) (Cai et al., 2024), and the integration of early exiting techniques with speculative sampling methodologies (Elhoushi et al., 2024). For token verification, researchers have predominantly employed three primary methods: greedy sampling, speculative sampling(Leviathan et al., 2023), and typical acceptance(Cai et al., 2024).

However, existing methods are limited to a **dualistic** relationship of cooperation between a draft model and a target model. The disparity in inference capabilities between these two models results in a small token average acceptance length, restricting the speedup ratio of speculative sampling. Although Chen et al. (2023b) propose cascading large and small models as the draft model, during

inference, it still utilizes a single draft model in conjunction with the target model. Meanwhile, existing works predominantly focus on direct algorithmic improvements, without conducting theoretical modeling specific to speculative decoding, resulting in a framework that lacks flexibility and controllability. Therefore, we have conducted theoretical modeling and analysis of existing speculative sampling methods. Building upon this foundation, we extend the concept of dualistic speculative decoding to **polybasic** speculative decoding. Specifically, our preliminary exploration revealed two key rules, laying the foundation for designing an efficient polybasic speculative decoding system. Firstly, we discovered that when the polybasic speculative decoding achieves optimal inference speed, there exists a significant correlation between the number of forward propagation executions for each model and the average token acceptance length between models. This finding enables us to calculate the ideal inference time for the polybasic speculative decoding system, providing a solid theoretical basis for subsequent research. Secondly, we conducted an in-depth study on the impact of speculative sampling on the performance of polybasic speculative decoding. The results indicate that introducing a carefully designed speculative sampling strategy can significantly improve the stability of token acceptance. This discovery not only optimizes system performance but also provides new insights into addressing uncertainty issues in polybasic speculative decoding.

Based on the aforementioned key insights, we synthesized a unified theoretical framework for polybasic speculative decoding, deriving the ideal inference time. This framework enables the evaluation of a model's potential to enhance inference speed through the calculation of its capabilities. According to this theory, we propose an innovative polybasic speculative decoding design method and have successfully implemented a specific design scheme. Through rigorous experimental validation, our method demonstrates significant performance advantages over dualistic speculative decoding, achieving higher acceleration ratios. To comprehensively evaluate system performance, we conducted extensive testing across a diverse set of tasks, including MT-bench(Zheng et al., 2023), translation, summarization, QA, math reasoning, and retrieval-augmented generation (RAG). The experimental results are encouraging: our system can increase inference speed to **3x-4x** that of the original model while maintaining output quality. The main contributions are summarized as follows:

- We provided a theoretical analysis for the ideal inference time in the polybasic speculative decoding system. We can use this analysis to determine whether adding a model to the system can improve inference speed.

- We theoretically elucidated the importance of speculative sampling in the polybasic decoding speculative systems. Our analysis demonstrated that speculative sampling plays a crucial role in stabilizing the average acceptance length between models, thereby enhancing the overall efficiency and reliability of the speculative decoding process.

- We designed polybasic speculative decoding, demonstrating both theoretically and experimentally that this system can significantly accelerate the inference of the target model. Furthermore, this method is orthogonal to most current speculative methods.

- Our method achieved remarkable latency speedup ratios of **3.31x-4.01x** for LLaMA2-Chat 7B, up to **3.87x** for LLaMA3-8B, and up to **4.43x** for Vicuna-7B. The output of the polybasic system aligns with the original model while maintaining the latency speedup ratios.

## 2 RELATED WORK

### 2.1 BACKGROUND

Speculative decoding has emerged as a prominent paradigm for accelerating inference in large language models. The field can be systematically categorized into two primary domains: drafting methodologies and verification techniques.

**Drafting Methodologies** Drafting approaches are bifurcated into independent and self-drafting strategies. Independent drafting employs distinct models for token generation, which can be either fine-tuned or tuning-free. Fine-tuned drafters, exemplified by SpecDec (Xia et al., 2023b) and BiLD (Kim et al., 2024), undergo task-specific optimization. Conversely, tuning-free drafters such as Speculative Decoding (Leviathan et al., 2023) and StagedSpec (Spector & Ré, 2023) utilize pre-existing models without additional training.

Self-drafting methodologies leverage the intrinsic architecture of the target model. These encompass FFN Heads approaches, including Blockwise (Stern et al., 2018) and Medusa (Cai et al., 2024); Early Exiting techniques, such as PPD (Yang et al., 2023) and Self-Speculative (Zhang & Chen, 2023); and Mask-Predict methods, exemplified by Parallel Decoding (Santilli et al., 2023) and Lookahead Decoding (Zhao et al., 2024).

**Verification Techniques**   Verification methods, crucial for maintaining the fidelity of drafted tokens, are categorized into three principal approaches. Greedy Decoding algorithms, both lossless and approximate, are represented by works such as SpecDec (Xia et al., 2023b) and BiLD (Kim et al., 2024). Speculative Sampling, introduced by Leviathan et al. (2023), offers both lossless and approximate variants, with notable extensions including DistillSpec (Zhou et al., 2023) and Online Speculative (Liu et al., 2023). The Token Tree Verification approach, as demonstrated by SpecInfer (Miao et al., 2024) and StagedSpec (Spector & Ré, 2023), presents an alternative verification paradigm.

## 2.2 PRELIMINARIES

Speculative decoding is characterized by accelerating LLM decoding while precisely maintaining the model's output distribution. We can introduce the process of dualistic speculative decoding based on the "draft-then-verify" paradigm.

**Drafting.**   Speculative decoding operates iteratively at each decoding step, efficiently generating multiple prospective tokens as a conjecture of the target LLM's output. More formally, given an input sequence $x_1, \ldots, x_t$ and a target LLM $\mathcal{M}_q$, this paradigm leverages an efficient draft model $\mathcal{M}_p$ to produce the subsequent $K$ drafted tokens:

$$p_1, \ldots, p_K = \text{DRAFT}\left(x_{\leq t}, \mathcal{M}_p\right),$$
$$\widetilde{x}_i \sim p_i, \quad i = 1, \ldots, K,$$

where $\text{DRAFT}(\cdot)$ denotes various drafting strategies, $p$ is the conditional probability distribution calculated by $\mathcal{M}_p$, and $\widetilde{x}_i$ denotes the drafted token sampled from $p_i$.

**Verification.**   Subsequently, the target LLM $\mathcal{M}_q$ performs parallel verification of these drafted tokens. Given the input sequence $x_1, \ldots, x_t$ and the draft $\widetilde{x}_1, \ldots, \widetilde{x}_K$, Speculative Decoding computes $K + 1$ probability distributions concurrently using $\mathcal{M}_q$:

$$q_i = \mathcal{M}_q\left(x \mid x_{\leq t}, \widetilde{x}_{<i}\right), \quad i = 1, \ldots, K+1.$$

Subsequently, each drafted token $\widetilde{x}_i$ undergoes verification through a specific criterion $\text{VERIFY}\left(\widetilde{x}_i, p_i, q_i\right)$. Only tokens satisfying this criterion are retained as final outputs, thereby ensuring consistency with the target LLM's quality standards. In the event of verification failure, the first non-compliant drafted token $\widetilde{x}_c$ is subject to correction via the strategy $\text{CORRECT}\left(p_c, q_c\right)$. To maintain output integrity, all drafted tokens subsequent to position $c$ are discarded. Conversely, if all tokens pass verification, an additional token $x_{t+K+1}$ is sampled from $q_{K+1}$ by:

$$x_{t+K+1} \sim q_{K+1} = \mathcal{M}_q\left(x \mid x_{\leq t+K}\right).$$

**Speculative sampling.**   Speculative sampling (Leviathan et al., 2023) is a method to sample from a target distribution $q(x)$ using an auxiliary distribution $p(x)$. We draw $x$ from $p(x)$ and accept it with probability $\min(1, \frac{q(x)}{p(x)})$. If rejected, we repeat the process. This is equivalent to accepting when $p(x) \leq q(x)$, and rejecting with probability $1 - \frac{q(x)}{p(x)}$ when $p(x) > q(x)$, drawing from $q'(x) = \text{norm}(\max(0, q(x) - p(x)))$ upon rejection. As proven in Appendix A.1 of speculative sampling, this method equates to sampling directly from the target LLM $\mathcal{M}_q$.

## 3 POLYBASIC SPECULATIVE DECODING

In this section, we will introduce our **polybasic speculative decoding** theory. Specifically, in Section 3.1, we provide a detailed exposition of our theoretical framework. In Section 3.2, we present the construction of polybasic speculative decoding along with its algorithmic workflow.

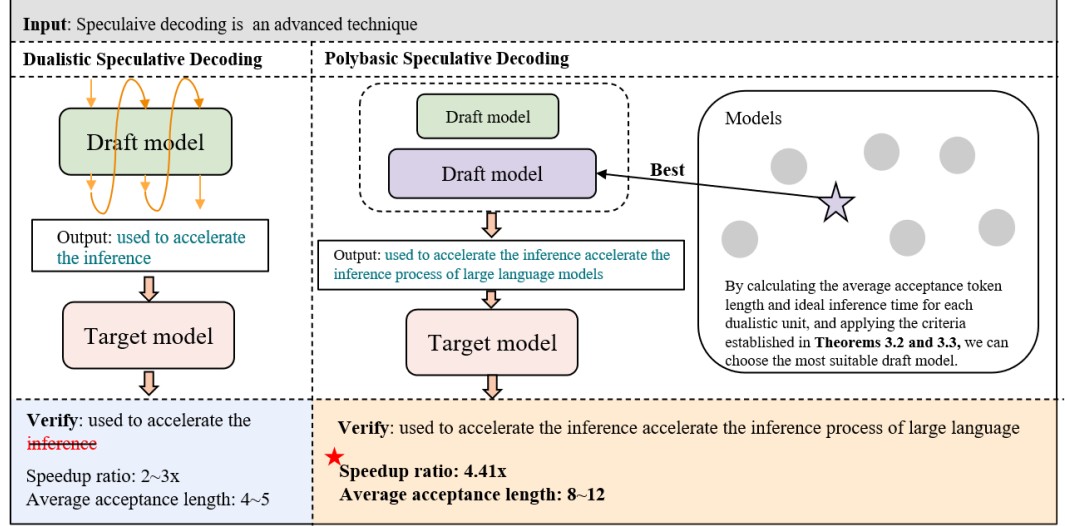

Figure 1: A comparison of the dualistic and polybasic speculative decoding. Our polybasic speculative decoding incorporates multiple draft models strategically selected based on Theorems 3.2 and 3.3, and achieve a **4.41**× speedup ratio and an improved average acceptance length of **8-12** tokens.

## 3.1 THEORETICAL FRAMEWORK

In Section 2.2, we delineated the algorithmic workflow of dualistic speculative decoding and conducted a comprehensive analysis. Through this analysis, we discerned that, to analyze the acceleration ratio of polybasic speculative decoding, it is essential to model the acceptance tokens length and the number of inference iterations between models. Therefore, we begin by postulating that the acceptance tokens length, denoted as $L$, can be characterized as a random variable following a Gaussian distribution with mean $\mu$ and variance $\sigma^2$, expressed as $L \sim \mathcal{N}(\mu, \sigma^2)$, where $\mathcal{N}(\mu, \sigma^2)$ represents the normal distribution.

For the convenience of discussion, we construct a polybasic speculative decoding system involving a sequence of models $\{M_i\}_{i=1}^n$, where models with higher inferential capacity and larger parameter counts serve as "target models" for their immediate successors. Specifically, for any $i \in \{1, \ldots, n-1\}$, model $M_i$ acts as the target model for $M_{i+1}$. The resulting "draft model", denoted as $D_i$, is composed of the models $(M_i, \ldots, M_n)$ and exhibits inferential capabilities more closely aligned with the next higher-level model $M_{i-1}$. This hierarchical structure can be formally expressed as $D_i = (M_i, \ldots, M_n)$, for $i \in \{1, \ldots, n-1\}$. This design principle aims to incrementally increase the token acceptance length of the entire system, denoted as $L_{D_i}$, such that $\mathbb{E}[L_{D_i}] > \mathbb{E}[L_{D_{i+1}}]$, for $i \in \{1, \ldots, n-2\}$ where $\mathbb{E}[\cdot]$ denotes the expected value operator. Then, to optimize the performance of our polybasic speculative decoding, we introduce the concept of ideal forward count, denoted as $\phi_i$ for model $M_i$, which represents the optimal number of forward passes required to generate tokens that are likely to be accepted by the previous model $M_{i-1}$. Through empirical analysis, we found that the system achieves its maximum acceleration ratio when the $\phi_i$ satisfies:

$$\phi_i = \begin{cases} \frac{N}{L_1} & \text{if } i = 1 \\ \frac{N}{L_i \cdot \left\lceil \frac{L_{i-1}}{L_i} \right\rceil} & \text{if } 1 < i < n \\ \alpha \cdot \phi_{n-1} & \text{if } i = n \end{cases}$$

where $N$ is the total number of tokens, $\alpha$ is a scaling factor related to the inferential capability of the smallest model $M_n$ and the specific speculative decoding method employed. To further analyze the ideal inference time of polybasic speculative decoding, we can first propose the lemma A.1

**Lemma 3.1.** *We can substitute $L$ with its expected value $\mathbb{E}[L]$.*

The rigorous proof of this substitution is provided in Appendix A.3. Combined with the $\phi_i$ and LemmaA.1, we can now express the ideal inference time $T$, which represents the theoretical optimal

inference time of our polybasic speculative decoding system:

$$T = T_{M_1} + T_{D_2}$$

$$= \phi_1 \cdot T_1 + \sum_{i=2}^{n} \phi_i \cdot T_i$$

$$= \sum_{i=1}^{n-1} \frac{N}{\mathbb{E}[L_i] \cdot \left\lceil \frac{\mathbb{E}[L_{i-1}]}{\mathbb{E}[L_i]} \right\rceil} \cdot T_i + \alpha \cdot \frac{N}{\mathbb{E}[L_{n-1}] \cdot \left\lceil \frac{\mathbb{E}[L_{n-2}]}{\mathbb{E}[L_{n-1}]} \right\rceil} \cdot T_n$$

where $T_i$ is the average inference time of the $i$-th model, and $\mathbb{E}[L_0] = 0$.

To facilitate the optimal selection of models for polybasic speculative decoding, we propose a set of design guidelines. To elucidate the efficacy of these guidelines, we extend our analysis from a two-model system to a three-model configuration, using this expansion as an illustrative example. Specifically, we propose Theorem 3.2, which serves as a foundational principle for our framework.

**Theorem 3.2.** *If either of the following conditions is satisfied:*

$$\frac{T_2'}{T_1} < 2\mathbb{E}[L_2]' \cdot \left( \frac{1}{\mathbb{E}[L_1]} - \frac{1}{\mathbb{E}[L_1]'} \right) \quad or \quad \frac{T_2'}{T_2} < \alpha \cdot \left( \frac{\mathbb{E}[L_1]}{2\mathbb{E}[L_2]'} - 1 \right)$$

*where $\mathbb{E}[L_1]' > \mathbb{E}[L_1]$ and $2\mathbb{E}[L_2]' > \mathbb{E}[L_1]$, then the total inference time of the three-model speculative decoding is less than the dualistic speculative decoding.*

*Proof.* For $i = 2$:

$$T = \frac{N}{\mathbb{E}[L_1]} \cdot T_1 + \alpha \cdot \frac{N}{\mathbb{E}[L_1]} \cdot T_2 \tag{1}$$

For $i = 3$:

$$T = \frac{N}{\mathbb{E}[L_1]'} \cdot T_1 + \frac{N}{\mathbb{E}[L_2]' \cdot \left\lceil \frac{\mathbb{E}[L_1]'}{\mathbb{E}[L_2]'} \right\rceil} \cdot T_2' + \alpha \cdot \frac{N}{\mathbb{E}[L_2]' \cdot \left\lceil \frac{\mathbb{E}[L_1]'}{\mathbb{E}[L_2]'} \right\rceil} \cdot T_3' \tag{2}$$

where $T_i$ is the inference time of the $i$-th model, $\alpha$ is considered to be equal in both equations, and $T_2 = T_3'$.

Because $\left\lceil \frac{\mathbb{E}[L_1]'}{\mathbb{E}[L_2]'} \right\rceil \geq 2$, we can calculate the difference between Equation 1 and Equation 2:

$$N \cdot \left( \frac{1}{\mathbb{E}[L_1]'} - \frac{1}{\mathbb{E}[L_1]} \right) \cdot T_1 + \frac{N}{2\mathbb{E}[L_2]'} \cdot T_2' + \alpha \cdot N \cdot \left( \frac{1}{2\mathbb{E}[L_2]'} - \frac{1}{\mathbb{E}[L_1]} \right) \cdot T_2 < 0$$

The expression is less than 0 if either of the following conditions is met:

Condition 1: Sum of the first two terms is less than 0

$$N \cdot \left( \frac{1}{\mathbb{E}[L_1]'} - \frac{1}{\mathbb{E}[L_1]} \right) \cdot T_1 + \frac{N}{2\mathbb{E}[L_2]'} \cdot T_2' < 0$$

$$\Leftrightarrow \frac{T_2'}{T_1} < 2\mathbb{E}[L_2]' \cdot \left( \frac{1}{\mathbb{E}[L_1]} - \frac{1}{\mathbb{E}[L_1]'} \right)$$

OR

Condition 2: Sum of the last two terms is less than 0

$$\frac{N}{2\mathbb{E}[L_2]'} \cdot T_2' + \alpha \cdot N \cdot \left( \frac{1}{2\mathbb{E}[L_2]'} - \frac{1}{\mathbb{E}[L_1]} \right) \cdot T_2 < 0$$

$$\Leftrightarrow \frac{T_2'}{T_2} < \alpha \cdot \left( \frac{\mathbb{E}[L_1]}{2\mathbb{E}[L_2]'} - 1 \right)$$

Therefore, the entire expression is less than 0 when either of the following inequalities is satisfied:

$$\frac{T_2'}{T_1} < 2\mathbb{E}[L_2]' \cdot \left( \frac{1}{\mathbb{E}[L_1]} - \frac{1}{\mathbb{E}[L_1]'} \right) \quad OR \quad \frac{T_2'}{T_2} < \alpha \cdot \left( \frac{\mathbb{E}[L_1]}{2\mathbb{E}[L_2]'} - 1 \right)$$

$$\square$$

This theorem provides a theoretical foundation for model selection in polybasic speculative decoding and establishes a basis for computing the ideal acceleration ratio. Then we use Theorem 3.2 to construct a polybasic speculative decoding model. However, we discovered instances of unstable acceptance token length, which affected the method's acceleration. Therefore, we conduct an analysis of the sampling method.

Specifically, we found that using speculative sampling can lead to more stable acceptance token length. By using speculative sampling, the number of tokens produced can be modeled as a capped geometric variable (Leviathan et al., 2023), with success probability $1 - \alpha$ and cap $n$.

$$\mu = \mathbb{E}[L] = \frac{1 - \alpha^{n+1}}{1 - \alpha} \tag{3}$$

where $\alpha$ represents the failure probability in each step, and $n$ is the maximum number of steps. The detailed derivation and proof of Equation 3 can be found in Appendix A.1. Building upon this definition, we proposed Theorem 3.3 during our comparative analysis of speculative sampling.

**Theorem 3.3.** *When the success probability $1 - \alpha$ is high, the acceptance token length exhibits very low relative variability.*

Having established the expected value $\mu$, we can employ a similar approach to calculate the variance $\sigma^2$ of the token generation process. The detailed derivation and proof for $\sigma^2$ are presented in Appendix A.2.

$$\sigma^2 = Var(L) = \frac{\alpha[1 - (n^2 - 1)\alpha^n] - (n^2 - 1)\alpha^{n+1}}{(1 - \alpha)^2}$$

Based on the expressions for $\mu$ and $\sigma^2$, we can now derive a measure of relative variability in our polybasic speculative decoding:

$$\frac{\sigma}{\mu} = \frac{\sqrt{\alpha[1 - (n^2 - 1)\alpha^n] - (n^2 - 1)\alpha^{n+1}}}{(1 - \alpha)(1 - \alpha^n)} \tag{4}$$

As $\alpha \to 0$, $\frac{\sigma}{\mu} \to 0$. This indicates that when the success probability is high (i.e., $1 - \alpha$ is high), the system exhibits very low relative variability (Appendix A.3). This means the token generation process becomes highly stable and predictable, thus supporting the conclusion that speculative sampling can effectively reduce variability in the polybasic speculative decoding. This stability contributes to improving the overall efficiency and performance of the system.

## 3.2 ALGORITHM

We propose a theoretical framework for polybasic speculative decoding, founded on the composition of dualistic speculative decoding units. This framework establishes a hierarchical structure of models, where combinations of varying model sizes yield draft models with enhanced inference capabilities. By calculating the average acceptance token length and ideal inference time for each dualistic unit, and applying the criteria established in Theorems 3.2 and 3.3, we can optimize the selection of dualistic processes to construct polybasic speculative decoding systems with superior acceleration ratios. This approach allows for the systematic design of more efficient large language model inference systems. Based on the framework, we propose a construction method for a polybasic speculative decoding that can reduce the inference time of the original dualistic system and improve the acceleration ratio.

First, we can select a suitable dualistic speculative decoding, such as EAGLE (Li et al., 2024a;b), SpS (Leviathan et al., 2023), etc. We choose EAGLE as the smallest draft model. EAGLE is a method that performs speculative sampling at the feature layer, achieving impressive inference acceleration.

Then, we selected a 4-bit quantization LLM as the intermediate model $M_2$. This choice is motivated by both Theorem 3.2 and Theorem 3.3. The 4-bit quantization LLM can maintain good accuracy

while achieving fast inference speeds after deployment. Through calculations presented in Table 1, we can verify that its post-processing time ($T_{\text{post}}$) is indeed less than the pre-processing time ($T_{\text{pre}}$) of the target model $M_1$, satisfying the necessary condition outlined in Theorem 3.2. Additionally, Theorem 3.3 suggests that the efficiency of speculative sampling is optimized when adjacent models have similar capabilities. In this case, we use AffineQuant (Ma et al., 2024) and OmniQuant (Shao et al., 2023) to quantize the target model $M_1$, ensuring that $M_1$ and $M_2$ have comparable capabilities while maintaining the performance advantages of the original model.

Finally, we use speculative sampling to ensure the stability of accepted tokens. This approach satisfies the necessary condition from Theorem 3.2 and aligns with the efficiency optimization principle from Theorem 3.3, potentially contributing to the overall acceleration and performance of the polybasic speculative decoding.

Table 1: Comparison of Single Model Performance ($T_i$) and Dualistic Model Metrics ($\mu_i$). Based on these , we can calculate and compare the $T$ values for both the dualistic and polybasic systems.

| Single Model | | Dualistic Model | |
| --- | --- | --- | --- |
| **Model** | $T_i$ | **Combination** | $\mu_i$ |
| $M_1$: Vicuna-7B | 25ms | $M_1 - M_2$ | 6.26 |
| $M_2$: Affinequant Quantized | 7ms | $M_1 - M_3$ | 4.34 |
| $M_3$: EAGLE | 4ms | $M_2 - M_3$ | 4.36 |

We present the algorithm1 of our polybasic speculative decoding.

## 4    EXPERIMENTS

**Models and tasks.** We conducted experiments on Vicuna-7B, LLaMA2-chat-7B, and LLaMA3-7B-Instruct. We evaluated our multi-model speculative system in SpecBench(Xia et al., 2024), across multiple tasks including multi-turn conversation, translation, summarization, question answering, mathematical reasoning, and retrieval-augmented generation, employing the MT-bench (Zheng et al., 2023), WMT14 DE-EN, CNN/Daily Mail (Nallapati et al., 2016), Natural Questions (Kwiatkowski et al., 2019), GSM8K (Cobbe et al., 2021), and DPR Karpukhin et al. (2020). Speculative sampling (Leviathan et al., 2023) conducted experiments with a batch size of 1, similarly, the majority of our experiments also adopted this setting.

**Metrics.** Like other speculative sampling-based methods, we assess acceleration effects using the following metrics:

- Walltime speedup ratio $c$ : The actual test speedup ratio relative to vanilla autoregressive decoding.

- Average acceptance length $\mu$ : The average number of tokens accepted per forward pass of the target LLM.

**Training and Quantization.** For training, we follow the setup outlined in EAGLE (Li et al., 2024a), conducting training on the ShareGPT dataset. We trained a corresponding draft model for both the target model and its respective quantized model. For quantization, we primarily use Affinequant (Ma et al., 2024) as our quantization method. We set both the weight quantization bits and activation quantization bits to 4, with a group size of 128. All our experiments, including training, inference, and the reproduction of EAGLE results, were conducted on NVIDIA A800 80G GPUs, ensuring consistent and comparable performance across all aspects of our study.

### 4.1    EFFECTIVENESS

Figures 2 and 3, along with Table 2, display the speedup ratios of our polybasic speculative decoding system. We have demonstrated that constructing polybasic speculative decodling system based on our two proposed claims can achieve superior acceleration compared to dualistic systems. In specialized categories such as MT-bench, Translation, QA, and Math, our approach consistently achieves

---

**Algorithm 1** Three-model Speculative Model

---

**Require:** Target language model $M_1$, draft model $M_2$ and $M_3$, input sequence $x_1, \ldots, x_n$, block size $K$, target sequence length $N$, drafting strategy DRAFT, verification criterion VERIFY, and correction strategy CORRECT;
1: **initialize** $cnt \leftarrow 0, m \leftarrow n$
2: **while** $n < N$ **do**
    *// Drafting: obtain distributions from $M_3$ efficiently*
3:    Set $p_1, \ldots, p_K \leftarrow$ DRAFT $(x_{\leq n}, M_3)$
    *// Drafting: sample $K$ drafted tokens*
4:    Sample $\widetilde{x}_i \sim p_i, i = 1, \ldots, K$
    *// Verification: compute $K + 1$ distributions in parallel*
5:    Set $q_i \leftarrow M_2 \left(x \mid x_{\leq n}, \widetilde{x}_{<i}\right), i = 1, \ldots, K + 1$
    *// Verification: verify each drafted token by $M_2$*
6:    **for** $i = 1 : K$ **do**
7:        **if** VERIFY $(\widetilde{x}_i, p_i, q_i)$ **then**
8:            Set $x_{n+i} \leftarrow \widetilde{x}_i$ and $n \leftarrow n + 1$
9:        **else**
10:          $x_{n+i} \leftarrow$ CORRECT $(p_i, q_i)$
11:          and Exit for loop.
12:        **end if**
13:    **end for**
14:    If all drafted tokens are accepted, sample next token $x_{n+1} \sim q_{K+1}$ and set $n \leftarrow n + 1$.
    *// Verification: verify each drafted token by $M_3$*
15:    **if** $cnt < \mu_1$ **then**
16:        $cnt \leftarrow cnt +$ accepted drafted tokens
17:        continue
18:    **else**
19:        Set $q_i \leftarrow M_3 \left(x \mid x_{\leq m}, \widetilde{x}_{<i}\right), i = 1, \ldots, cnt + 1$
20:        **for** $i = 1 : cnt$ **do**
21:            **if** VERIFY $(\widetilde{x}_i, p_i, q_i)$ **then**
22:               Set $x_{m+i} \leftarrow \widetilde{x}_i$ and $m \leftarrow m + 1$
23:            **else**
24:               $x_{m+i} \leftarrow$ CORRECT $(p_i, q_i)$
25:               and Exit for loop.
26:            **end if**
27:        **end for**
28:        $n \leftarrow m$
29:        If all drafted tokens are accepted, sample next token $x_{m+1} \sim q_{cnt+1}$ and set $n \leftarrow m + 1$.
30:    **end if**
31: **end while**

---

over 3x acceleration, with notable peaks in performance. The LlaMA2chat 7B model attains a $4.10\times$ acceleration in the MT-bench, while the Vicuna 7B model reaches an impressive $4.43\times$ acceleration in the Math task. These task-specific results represent substantial improvements over existing techniques like EAGLE, which typically achieve acceleration ratios between $2\times$ and $3\times$. Overall, our method maintains strong acceleration ratios above $3\times$ for all tested models ($3.16\times$ for Vicuna 7B, $3.31\times$ for LlaMA3 8B Instruct, and $3.66\times$ for LlaMA2chat 7B). This consistent performance across varied tasks and models underscores the versatility and effectiveness of our polybasic speculative decoding system.

As show in Table 2, our method demonstrates remarkable efficiency through significantly increased average acceptance lengths across all tasks. We approach consistently achieves average acceptance lengths above 9.4 tokens, with LlaMA2chat 7B model showcasing exceptional performance. This model reaches an impressive average acceptance length of 10.47 tokens in the MT-bench and maintains high efficiency across other tasks, with an overall average of 9.84 tokens. These acceptance lengths significantly surpass those of existing speculative sampling methods.

## 4.2 ABLATION STUDY

To investigate the impact of speculative sampling and greedy sampling on the stability of average acceptance length in our multi-tier system, we conducted an ablation study. We randomly selected

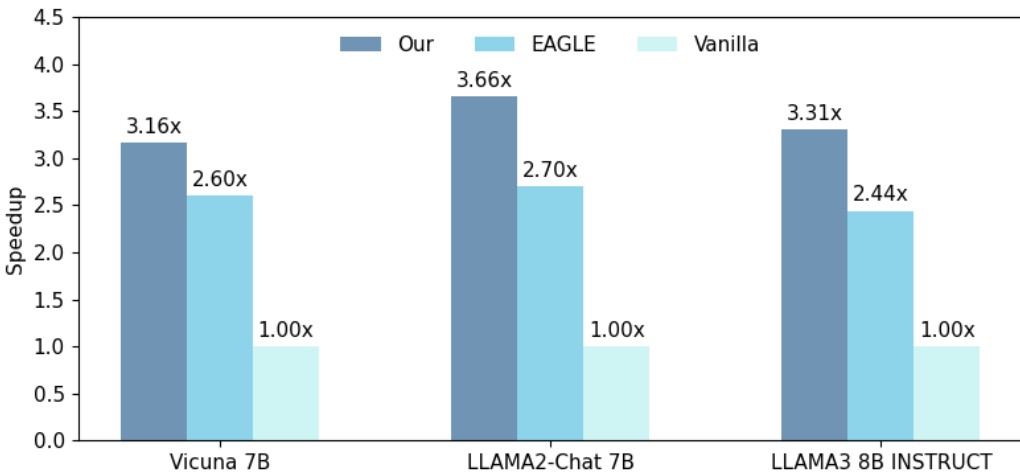

Figure 2: Speedup ratio of Vicuna, LLaMA2-Chat and LLaMA3 Instruct inference latency on the Spec-Bench. Our approach consistently achieves the highest speedup ratios, ranging from **3.16×** to an impressive **3.66×**, significantly outperforming both the EAGLE method and the vanilla baseline. The consistent outperformance over existing methods, culminating in the **highest** overall speedup on the Spec-Bench.

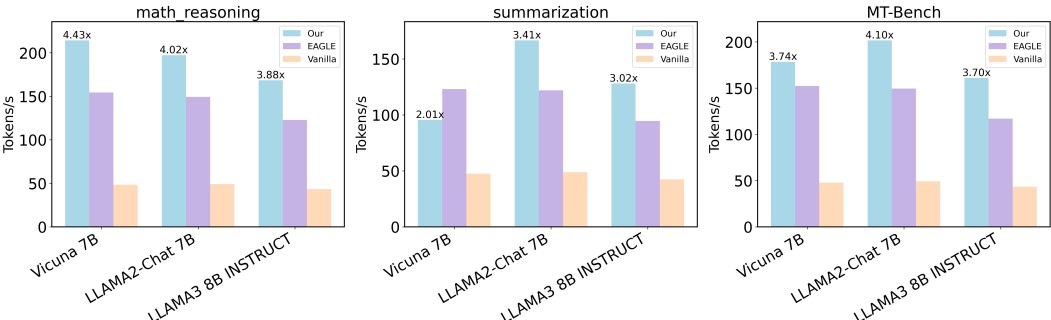

Figure 3: Performance Comparison across different tasks. Our method demonstrates its peak performance in the math task, achieving an impressive **4.43×** speedup with the Vicuna 7B model.

50 questions and applied both sampling methods to generate acceptance length lists. To visualize the results, we plotted the variances of these two datasets, as shown in the figure 4.

The graph clearly demonstrates that the speculative sampling method exhibits significantly lower variance compared to the greedy sampling method. This indicates that speculative sampling produces more consistent and stable acceptance lengths across different queries. In contrast, greedy sampling shows higher variance, implying greater fluctuations in acceptance lengths between queries. These findings highlight the advantage of speculative sampling in maintaining the stability of our polybasic system's performance.

### 4.3 LIMITATIONS AND DISCUSS

In dualistic speculative decoding systems, the KV cache size grows linearly with text length, presenting a critical bottleneck for inference acceleration. This challenge similarly applies to our polybasic speculative decoding system. As show in Figure 3 and the table 2, acceleration ratios for RAG and summarization tasks are notably lower compared to other tasks. Therefore, while implementing our two claims to construct a polybasic speculative decoding system, it is crucial to consider the KV cache issues introduced by incorporating additional models(Xiao et al., 2023)(Zhang et al.,

Table 2: Average acceptance length and speedup ratio on different tasks

|  | Model | MT | | Trans. | | Sum. | | QA | |
|---|---|---|---|---|---|---|---|---|---|
|  | | $c$ | $\mu$ | $c$ | $\mu$ | $c$ | $\mu$ | $c$ | $\mu$ |
| Our | Vicuna 7B | **3.77x** | **11.22** | **3.07x** | **7.76** | 2.01x | **10.18** | **3.65x** | **9.53** |
| | LlaMA3 8B Instruct | **3.70x** | **9.97** | **3.39x** | **8.86** | **3.02x** | **9.38** | **3.16x** | **9.08** |
| | LlaMA2chat 7B | **4.10x** | **10.47** | **3.46x** | **9.15** | **3.41x** | **9.86** | **3.61x** | **9.49** |
| EAGLE | Vicuna 7B | 3.19x | 4.76 | 2.07x | 3.22 | 2.59x | 3.96 | 2.45x | 3.71 |
| | LlaMA3 8B | 2.69x | 3.99 | 2.37x | 3.53 | 2.23x | 3.58 | 2.21x | 3.42 |
| | LlaMA2chat 7B | 3.04x | 4.48 | 2.61x | 3.96 | 2.50x | 4.04 | 2.55x | 4.05 |

|  | Model | Math | | RAG | | Overall | |
|---|---|---|---|---|---|---|---|
|  | | $c$ | $\mu$ | $c$ | $\mu$ | $c$ | $\mu$ |
| Our | Vicuna 7B | **4.43x** | **10.28** | 1.78x | **10.31** | **3.16x** | **9.88** |
| | LlaMA3 8B | **3.87x** | **10.08** | **2.71x** | **9.24** | **3.31x** | **9.44** |
| | LlaMA2chat 7B | **4.02x** | **9.99** | **3.31x** | **10.08** | **3.66x** | **9.84** |
| EAGLE | Vicuna 7B | 3.19x | 4.72 | 2.15x | 3.95 | 2.61x | 4.34 |
| | LlaMA3 8B | 2.83x | 4.20 | 2.23x | 3.95 | 2.44x | 3.82 |
| | LlaMA2chat 7B | 3.04x | 4.68 | 2.40x | 4.19 | 2.70x | 4.30 |

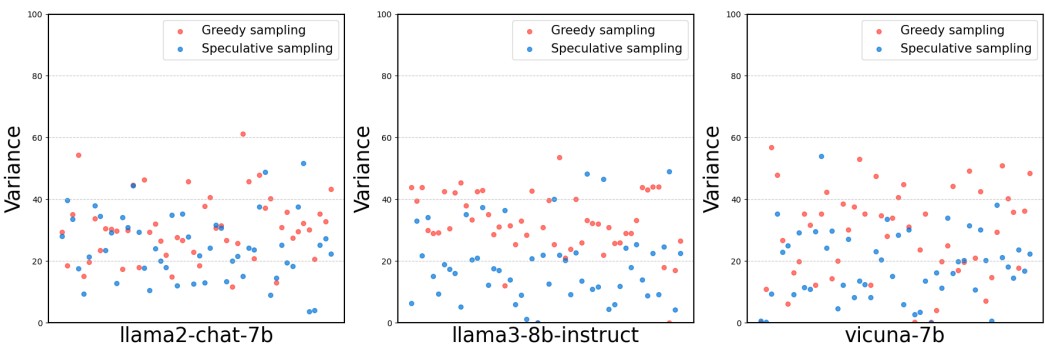

Figure 4: Variance Comparison of Greedy vs. Speculative Sampling.

2024b)(Zhang et al., 2024a)(Jin et al., 2024)(Jiang et al., 2023)(Ge et al., 2023). We plan to conduct further research on this aspect in our future work.

## 5 CONCLUSION

In this papaer, we introduce the polybasic speculative decoding system, an efficient framework for speculative sampling. Within this framework, we deduce a theorem to control the ideal inference time of speculative decoding systems. And we theoretically demonstrate the benefits of speculative sampling for enhancing the stability of average token acceptance length in polybasic speculative systems. We conducted extensive evaluations using various LLMs across Spec-Bench with multiple datasets. In our experiments, we achieved the highest average token acceptance and substantial speedup ratios.

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

# A STATISTICAL ANALYSIS OF ACCEPTANCE LENGTH

## A.1 CALCULATION OF MEAN ACCEPTANCE LENGTH

Given a geometric distribution truncated after $n$ trials, where the probability of success is $p = 1 - \alpha$.

We want to calculate:

$$S = \sum_{k=1}^{n-1} k \cdot (1-p)^{k-1}$$

Using the method of differences:

1. Define:

$$T = \sum_{k=1}^{n-1} (1-p)^{k-1} = \frac{1 - (1-p)^{n-1}}{p}$$

2. Calculate $S$ using shifted difference.

Consider the series:

$$S = 1 + 2(1-p) + 3(1-p)^2 + \cdots + (n-1)(1-p)^{n-2}$$

$$(1-p)S = (1-p) + 2(1-p)^2 + 3(1-p)^3 + \cdots + (n-1)(1-p)^{n-1}$$

Subtract the two equations:

$$S - (1-p)S = 1 + (1-p) + (1-p)^2 + \cdots + (1-p)^{n-2} - (n-1)(1-p)^{n-1}$$

$$pS = T - (n-1)(1-p)^{n-1}$$

3. Substitute $T$:

$$pS = \frac{1 - (1-p)^{n-1}}{p} - (n-1)(1-p)^{n-1}$$

$$S = \frac{1 - (1-p)^{n-1} - n(1-p)^{n-1} + (1-p)^n}{p^2}$$

The expectation $E(N)$ is:

$$E(N) = \sum_{k=1}^{n-1} k \cdot p \cdot (1-p)^{k-1} + n \cdot (1-p)^{n-1}$$

Substitute for $S$:

$$E(N) = \frac{1 - n(1-p)^{n-1} + (n-1)(1-p)^n}{p} + n \cdot (1-p)^{n-1}$$

Simplifying:

$$E(N) = \frac{1 - (1-p)^n}{p}$$

This formula gives the expected number of trials until success, assuming the $n$-th trial is successful.

A.2  CALCULATION OF VARIANCE IN ACCEPTANCE LENGTH

We begin by recalling the expectation of this distribution:

$$E(N) = \sum_{k=1}^{n-1} k \cdot p \cdot (1-p)^{k-1} + n \cdot (1-p)^{n-1} = \frac{1 - (1-p)^n}{p}$$

To derive the variance, we need to calculate $E(N^2)$. Let's define:

$$E(N^2) = \sum_{k=1}^{n-1} k^2 \cdot p \cdot (1-p)^{k-1} + n^2 \cdot (1-p)^{n-1}$$

To simplify our calculations, we introduce an auxiliary sum:

$$S = \sum_{k=1}^{n-1} k^2 \cdot (1-p)^{k-1}$$

We can now apply the method of differences:

$$S = 1 + 4(1-p) + 9(1-p)^2 + \cdots + (n-1)^2(1-p)^{n-2}$$
$$(1-p)S = (1-p) + 4(1-p)^2 + 9(1-p)^3 + \cdots + (n-1)^2(1-p)^{n-1}$$

Subtracting these equations yields:

$$pS = 1 + 3(1-p) + 5(1-p)^2 + \cdots + (2n-3)(1-p)^{n-2} - (n-1)^2(1-p)^{n-1}$$

We can further simplify this expression by splitting the sum and recognizing geometric series:

$$\begin{aligned} pS = {} & [1 + (1-p) + (1-p)^2 + \cdots + (1-p)^{n-2}] \\ & + [2(1-p) + 4(1-p)^2 + \cdots + (2n-4)(1-p)^{n-2}] \\ & - (n-1)^2(1-p)^{n-1} \end{aligned}$$

This simplifies to:

$$pS = \frac{1 - (1-p)^{n-1}}{p} + 2(1-p)\frac{1 - (1-p)^{n-2}}{p} - (n-1)^2(1-p)^{n-1}$$

Further algebraic manipulation leads to:

$$S = \frac{1 - (1-p)^{n-1}}{p^2} + \frac{2(1-p)[1 - (1-p)^{n-2}]}{p^2} - \frac{(n-1)^2(1-p)^{n-1}}{p}$$

Substituting back into the expression for $E(N^2)$:

$$E(N^2) = pS + n^2(1-p)^{n-1}$$

We arrive at the final expression for $E(N^2)$:

$$E(N^2) = \frac{1 - (1-p)^n(n^2 + 2n - 1) + 2(1-p)^{n+1}(n-1)}{p^2}$$

Now we can calculate the variance using the formula $Var(N) = E(N^2) - [E(N)]^2$:

$$
\begin{aligned}
Var(N) &= E(N^2) - [E(N)]^2 \\
&= \frac{1 - (1-p)^n(n^2 + 2n - 1) + 2(1-p)^{n+1}(n-1)}{p^2} - \left[\frac{1 - (1-p)^n}{p}\right]^2
\end{aligned}
$$

After simplification, we obtain the final expression for the variance:

$$Var(N) = \frac{(1-p)[1 - (1-p)^n(n^2 - 1)] - (1-p)^{n+1}(n^2 - 1)}{p^2}$$

This formula provides the variance of the truncated geometric distribution in terms of the success probability $p$ and the truncation point $n$.

### A.3 ANALYSIS OF ACCEPTANCE TOKEN LENGTH

**Lemma A.1.** *We can substitute $L$ with its expected value $\mathbb{E}[L]$.*

To analyze the ideal forward count in our polybasic speculative decoding, we introduce a probabilistic framework to account for the variability in token generation across different models. Let $L_i$ be a random variable representing the number of tokens generated by the model, with $\mathbb{E}[L_i] = \mu_i$ and $\mathrm{Var}(L_i) = \sigma_i^2$.

We focus on the term $1/L_i$, which is a critical component influencing the $\phi_i$ value. To analyze this term, we apply a second-order Taylor series expansion of the function $f(L_i) = 1/L_i$ around $\mu_i$:

$$f(L_i) \approx f(\mu_i) + f'(\mu_i)(L_i - \mu_i) + \frac{1}{2}f''(\mu_i)(L_i - \mu_i)^2$$

where $f(\mu_i) = 1/\mu_i$, $f'(\mu_i) = -1/\mu_i^2$, and $f''(\mu_i) = 2/\mu_i^3$.

Taking the expectation of the expanded function, we obtain:

$$\mathbb{E}[f(L_i)] \approx \frac{1}{\mu_i} - \frac{1}{\mu_i^2}\mathbb{E}[L_i - \mu_i] + \frac{1}{\mu_i^3}\mathbb{E}[(L_i - \mu_i)^2]$$

Given that $\mathbb{E}[L_i - \mu_i] = 0$ and $\mathbb{E}[(L_i - \mu_i)^2] = \sigma_i^2$, we arrive at:

$$\mathbb{E}[f(L_i)] \approx \frac{1}{\mu_i} + \frac{\sigma_i^2}{\mu_i^3}$$

The term $\sigma_i^2/\mu_i^3$ represents the additional expected value of $1/L_i$ due to the variability of $L_i$. The significance of this term depends on the relative magnitude of the variance $\sigma_i^2$ compared to the square

of the mean $\mu_i^2$. If $\sigma_{i}^2 \ll \mu_i^2$, indicating that the variability of $L_i$ is small relative to its expected value, then the $\sigma_i^2/\mu_i^3$ term becomes negligible compared to $1/\mu_i$. This observation provides a basis for potential simplification of our model in cases where the variability of $L_i$ is sufficiently low relative to its mean.

This analysis demonstrates that $\mathbb{E}[1/L_i] \approx 1/\mathbb{E}[L_i]$ when the coefficient of variation is small. Consequently, we can substitute $L$ with its expected value $\mathbb{E}[L]$ in the ideal inference time equation without significant loss of accuracy, as the effect of variability becomes negligible under these conditions.

