# OpenReview forum: "Polybasic Speculative Decoding Under a Theoretical Perspective"
_ICLR.cc/2025/Conference — ICLR 2025 Conference Withdrawn Submission_

### Official Review · Reviewer_3H2r · 2024-11-03

**Soundness:** 2
**Presentation:** 1
**Contribution:** 2
**Rating:** 3
**Confidence:** 3

**Summary:**

This paper aims to improve the speculative decoding (SD) process by generalizing the original SD framework that involves two models in the role of the draft and target model, respectively. This paper focused on leveraging multiple draft models to improve the speedup realized by the SD process. The paper claims to develop a solid theoretical framework to design the proposed polybasic SD with multiple draft models. It showcases significant speed-up on various benchmarks, including MT-bench, mathematical reasoning, summarization, translation, question-answering, and retrieval-augmented generation tasks.

**Strengths:**

1) The paper aims to improve the existing SD framework by utilizing multiple models as potential draft models.
2) The paper aspires to develop a theoretical foundation for SD and utilize it to guide the design of the polybasic SD method.
3) On a wide range of tasks, the proposed polybasic SD method showcases impressive inference speedup.

**Weaknesses:**

Given that the paper claims the theoretical foundation for SD as one of their key contributions, the reviewer found the presentation and results in the paper somewhat underwhelming. Improving the presentation of existing results by properly setting up notation, formalizing theoretical statements, and rigorously justifying various assumptions is essential to improve the quality of a future submission. Some of the comments in this direction are as follows:

1) The authors may want to first provide a clear and brief outline of what they want to establish via their theoretical analysis before stating their results.
2) Section 3.1 starts by assuming that accepted token lengths follow a Gaussian distribution. What is the justification for this? Is this done only to make the analysis simpler? Without a rigorous justification, it is not clear if the rest of the analysis provides any realistic insight. Could authors assume this distribution to be a more plausible discrete distribution?
3) The authors state that by empirical analysis they obtain a precise formulation for  $\phi_i$. There are no details provided regarding this analysis. If the results heavily rely on this precise formulation, it needs to be further explained and justified.
4) Please formally define *acceleration ratio* in Section 3.1.
5) Please define what $T_{M_1}$ and $T_{D_2}$ represent in line 218. Similarly, please define what $T_i$ vs. $T'_i$ and $L_i$ vs $L'_i$ are in Theorem 3.2
6) In Line 273, the authors talk about "...unstable acceptance token lengths...", which is not formally defined so far.
7) Please make the main result of Theorem 3.3 formal. In its present form, the theorem makes a qualitative statement that needs to be formalized.

The description of the main algorithm is not clear enough. For instance, Algorithm 1 does not seem to be utilizing $M_1$ anywhere.

The paper has multiple typos. Please consider performing a thorough proofread to fix the typos similar to the following:

1) In the abstract (line 18), `..is then serve` --> `...then serve`? Also, please change `theorem` to `theorems` Or `serve` to `serves`.
2) Line 189,  the *acceptance* token length --> the *accepted* token length?
3) Please consistently use *lemma* or *Lemma* throughout the text.
4) Line 423,  'We approach consistently...' --> 'Our approach consistently....'?
5) Section 4.3 title: 'Limitation and discuss' --> 'Limitation and discussion'?
6) Line 483, 'As show in Figure....' --> 'As shown in Figure...'?

**Questions:**

Please see the weaknesses section above. In addition, the reviewer has the following questions:

Line 53 discusses Chen et al. (2023b) and mentions that they only utilize a single draft model in conjunction with the target model. Looking at Figure 1 of https://arxiv.org/pdf/2312.11462, they do seem to be considering a horizontal cascade where different draft models are utilized with a single target model. Could you please expand your discussion of the related work and comment on this?

Line 330 states ".....ensuring that $M_1$ and $M_2$ have comparable capabilities while maintaining the performance advantage of the original model". Could the authors clarify what they mean here? Are we trying to deliberately lower the quality of $M_1$ so that it's comparable to $M_2$?

---

### Official Review · Reviewer_dMxT · 2024-11-04

**Soundness:** 1
**Presentation:** 1
**Contribution:** 2
**Rating:** 3
**Confidence:** 3

**Summary:**

**Background**: Speculative decoding refers to a suite of techniques to allow fast inference of LLMs. A common form of speculative decoding is based on the draft-and-verify (or more intuitively “propose-then-accept”) framework where a small, “drafter” model is used to generate next tokens, which are then verified in parallel by the large, target LLM of interest.

This paper proposes the so-called “polybasic speculative decoding”, a framework which relies on a chain of n LLMs ordered in decreasing sizes. When the i-th model is used as the target LLM, the first i-1 models are used together to form a drafter. Theoretically, the paper contributes several theorems to characterize the expected acceptance rate of the new framework. Empirically, the paper shows on standard benchmarks that the new framework allows 3-4 times latency speedup compared to the vanilla setting (i.e., directly sampling from the target LLM) .

**Strengths:**

While the idea of using multiple drafting models is not entirely new, the attempt of arranging n LLMs in a chain and constructing a draft-and-verify pipeline on them appears to be interesting. Though, the paper ends up studying the case of n=3 (see Algo 1). Empirical results verifying that the new approach provides 3-4 times inference speedup is encouraging.

**Weaknesses:**

Overall writing could be improved. More importantly, many important mathematical statements are not precisely stated. This is a major concern. Many statements are imprecise to the point of hindering understanding of the work. For instance, Lemma 3.1 has only one sentence that states “We can substitute L with its expected value E[L].”. I was left wondering “Substitute into where? What assumptions are made?” There is no objective function written until Lemma 3.1 that would allow me to substitute L with E[L]. I guess that perhaps Lemma 3.1 is meant for  $\phi_i$ (L207). This then leads to another trouble which is that the definition of $\phi_i$ is never justified beyond the phrase “Through empirical analysis”. No further detail is given. There are other examples. More  specific questions will be given in the Questions section of this form. Significance and in fact correctness of many statements are not easy to verify because of their impreciseness.

**Questions:**

**Major comments/questions:**

1. Clarify throughout that all the theoretical analysis is independent of any prefix. For instance, the random variable L is meant to denote the length of accepted tokens *on average* (independently of any input sequence). This is never stated. At L190, it is assumed that $L \sim \mathcal{N}(\mu, \sigma^2)$. Could you please provide a justification for why this assumption makes sense? L is a discrete variable. A normally distributed variable can take a negative value. More importantly, where is this normality assumption used in the paper?

2. Sec 3.1, L202. “we introduce the concept of ideal forward count, denoted as $\phi_i$ for model $M_i$ , which represents the optimal number of forward passes required to generate tokens.” This is arguably one of the most important parts in the paper. What does “number of forward passes“ mean? Also, “to generate tokens” at what granularity (e.g., for one speculative decoding block, for one query, for the whole test set)?

3. L203. The paper defines $\phi_i$ “through empirical analysis”. What is the empirical analysis? How did you come up with $\phi_i$ as defined in Sec 3.1? All further theoretical analysis appears to rely on this definition. But the definition itself is not fully justified.

4. Lemma 3.1: “We can substitute L with its expected value E[L].” Substitute L into where? What are the assumptions used for this Lemma?  The proof in Sec A.3 mentions “small coefficient of variation”. If this is needed, please explicitly state it as an assumption in the Lemma.

5. L217. Inference time is defined as $T = T_{M_1} + T_{D_2}$. Could you please define  $T_{M_1}$ and $T_{D_2}$? What is the unit of time? Is the inference time for one input sequence, or for the whole test set?

6. Theorem 3.2: What is the meaning of the two conditions? What are their practical implications?

7. Theorem 3.3 needs to be more precise. “the acceptance token length exhibits very low relative variability.“ Could you please quantify “very low” and define “relative variability”? Is Theorem 3.3 proved?

8. Figure 4: What does x-axis represent?


**Important to fix but less severe than the major concerns above:**


1. The words “polybasic” and “dualistic” in the introduction are never defined. I think directly defining them and mentioning pros and cons in the introduction would be helpful. The current introduction does not tell the reader what the proposal is.


2. L155: *“As proven in Appendix A.1 of speculative sampling, this method (speculative decoding) equates to sampling directly from the target LLM M_q.”* A.1 is proving an entirely different statement. Specifically it proves the mean of a truncated geometric distribution.

---

### Official Review · Reviewer_Edvg · 2024-11-04

**Soundness:** 2
**Presentation:** 2
**Contribution:** 2
**Rating:** 3
**Confidence:** 4

**Summary:**

The paper propose a polybasic speculative decoding framework supported by a solid theoretical foundation.  The method achieves latency speedup ratios of 3.31×-4.01× for LLaMA2-Chat 7B, up to 3.87× for LLaMA3-8B, and up to 4.43× for Vicuna-7B.

**Strengths:**

The empirical results are strong. The paper conducted experiments on 3 different models under a variety of settings, and showed 3-4x speedup, which surpasses the baseline method that only uses single draft models.

**Weaknesses:**

The theoretical parts are not sound:

- The concept "ideal forward count" $\phi_i$ is not properly defined. "Optimal number of forward passes required to generate tokens that are likely to be accepted by the previous model Mi−1". Is $\phi_i$ a random variable? What do you mean by "optimal"? Is $\phi_i$ obtained by optimizing some objective? What is "likely to be accepted"? Is this a high probability definition?
- Line 205-209 also comes from a contingent source of evidence: the definition of $\phi_i$ shouldn't be dependent on your specific empirical setting.
- The statment of Lemma 3.1 is not rigorous. The proof is not rigorous as well. In the proof, the authors neglect the randomness of $L_i$ by a extremely vague argument "when the variability of Li is sufficiently low relative to its mean, we can substitute L with its expected value E[L]". Instead, a proper development of lemma 3.1 will involve certain form of concentration inequalty, and the authors should quantify the errors of Talyor expansion and dropping the higher-order terms.
- In Theorem 3.2, the definition of $T_2'$ and $L_2'$ should be clearly stated in the theorem statement.
- Why would we need $\mathbb{E}[L_{D_i}] > \mathbb{E}[L_{D_{i+1}}]$ in the first place? There is no theoretical result supporting this goal.
- Line 275-277 the claim "we found that using speculative sampling can lead to more stable acceptance token length." is weird.


The novelty of the empirical algorithm is limited. Cascaded speculative decoding idea is not new and there are already multiple previous papers, including but not limit to:

- [1]. Cascade Speculative Drafting for Even Faster LLM Inference (https://arxiv.org/pdf/2312.11462)
- [2]. Accelerating LLM Inference with Staged Speculative Decoding (https://arxiv.org/pdf/2308.04623)

The theory developed in the paper can extend to speculative decoding systems with more than 4 models, but the experiments only restrict to 3-model cases.


The theoretical part seems to be disconnected with the empirical part. The results of Theorem 3.2  is not used in guiding the determination of the number of models $n$ and the selection of models $M_1,M_2,\dots, M_n$. E.g., I would expect the authors construct a pool of models and calculate the $T_i$ and $L_i$ as per Theorem 3.2 for each pair of the models. Besides, Theorem 3.3 is a simple characteristic of capped geometry variable and is a bit misleading (we want draft models with high acceptance rate instead of low acceptance length variance.)

**Questions:**

The acceptance length should be the property of the target distribution and the draft distribution. Consider $M_1$ as the target model and $M_2$ as the draft model. Then using $M_3$ to speed up $M_2$ with speculative decoding shouldn't modify the distribution of $M_2$, i.e., the generated tokens have the same distribution as it was generated by $M_2$ only. If the draft distribution of $M_2$ is the same as $(M_2, M_3)$, then the average acceptance length shouldn't change as well. Can you clarify if I misunderstand the terms? What is the reason behind $\mu$'s change in Table 2 for "ours" and "eagle"? Are the draft models different for "ours" and "eagle"?

---

### Official Review · Reviewer_QRqe · 2024-11-08

**Soundness:** 2
**Presentation:** 1
**Contribution:** 3
**Rating:** 3
**Confidence:** 4

**Summary:**

The authors propose speculative decoding where they use multiple models to draft and verify. The idea has theoretical backing to understand where multiple models make sense compared to existing setup of dualistic speculative decoding. The authors show impressive speedups over EAGLE.

“there exists a significant correlation between the number of forward propagation executions for each model and the average token acceptance length between models.” -> Interesting observation, but at some level why is this not intuitive, probabilisticly speaking more draft models  are run, higher the likelihood.

“Through empirical analysis, we found that the system achieves its maximum acceleration ratio when the φi satisfies:” -> Please link the figure for this, all your proofs depend on this.

“Figure 3: Performance Comparison across different tasks. Our method demonstrates its peak performance in the math task, achieving an impressive 4.43× speedup with the Vicuna 7B model.” -> Can authors comment on why in a certain case in EAGLE is faster ?

How will your system work with different paradigms which construct token tree  like SpecInfer or Medusa.

Further how will you handle ideas like Self-Speculative decoding, where intermediate layers are used as exit points to perform speculative decoding.

“When the success probability 1 − α is high” -> How do you calculate the success probability at inference ?

I am also curious how does this work compare to works like SpecInfer which builts multiple models to build a token tree verifier.

**Strengths:**

I think the speedups reported are quite impressive.

The limitations section is very well written and highlights the problems with implementation.

**Weaknesses:**

See Summary.

If the authors are able to clarify some of the questions and fix some of the writing. I am happy to bump up the score to a 5.

**Questions:**

See Summary.

---

### Note · Authors · 2024-11-25

I have read and agree with the venue's withdrawal policy on behalf of myself and my co-authors.